# idopNetwork Analysis of Salt-Responsive Transcriptomes Reveals Hub Regulatory Modules and Genes in *Populus euphratica*

**DOI:** 10.3390/ijms26094091

**Published:** 2025-04-25

**Authors:** Shuang Wu, Wenqi Pan, Ang Dong

**Affiliations:** 1Center for Computational Biology, School of Grassland Science, Beijing Forestry University, Beijing 100083, China; shuangwu@bimsa.cn (S.W.); wqpan1015@bjfu.edu.cn (W.P.); 2Beijing Key Laboratory of Topological Statistics and Applications for Complex Systems, Beijing Institute of Mathematical Sciences and Applications, Beijing 101408, China

**Keywords:** gene regulatory network, transcriptomics, salt response, Euphrates poplar, ordinary differential equations

## Abstract

Euphrates poplar (*Populus euphratica*) is known as a system model to study the genomic mechanisms underlying the salt resistance of woody species. To characterize how dynamic gene regulatory networks (GRNs) drive the defense response of this species to salt stress, we performed mRNA sequencing of *P. euphratica* roots under short-term (ST) and long-term (LT) salt stress treatments across multiple time points. Comparisons of these transcriptomes revealed the diverged gene expression patterns between the ST and LT treated samples. Based on the informative, dynamic, omnidirectional, and personalized networks model (idopNetwork), inter- and intra-module networks were constructed across different time points for both the ST and LT groups. Through the analysis of the inter-module network, we identified module 4 as the hub, containing the largest number of genes. Further analysis of the gene network within module 4 revealed that gene *XM_011048240.1* had the most prominent interactions with other genes. Under short-term salt stress, gene interactions within the network were predominantly promoted, whereas under long-term stress, these interactions shifted towards inhibition. As for the gene ontology (GO) annotation of differentially expressed genes, the results suggest that *P. euphratica* may employ distinct response mechanisms during the early and late stages of salt stress. Taking together, these results offer valuable insights into the regulatory mechanism involved in *P. euphratica*’s stress response, advancing our understanding of complex biological processes.

## 1. Introduction

Soil salinity, as a major abiotic stress, primarily arouses hyperosmotic stress and disrupts ion homeostasis in plant and then results in secondary stresses such as oxidative damage, leading to molecular damage, growth arrest, and even death [1,2]. The desert poplar, Euphrates poplar (*Populus euphratica*), has a wide natural distribution, ranging from North Africa, across the Middle East and Central Asia to western China. It can grow very well in regions with a wide variety of detrimental environmental conditions, such as the high salt content in soil and low or high temperatures in the air [3]. Previous studies have shown that regenerated *P. euphratica* can survive under hydroponic conditions with 450 mM NaCl [4]. Therefore, *P. euphratica* is widely considered as an ideal model for elucidating the mechanisms of salt tolerance in woody species [5,6].

*Populus trichocarpa*, a widely recognized model tree, is a perennial woody plant known for its rapid growth and high yield, making it a valuable species in ecology [7]. However, previous studies have shown that *P. euphratica*, as a species similar to *P. trichocarpa*, has greater advantages in the study of salt tolerance mechanisms [5]. Despite this, research on salt tolerance genes in *P. euphratica* remains incomplete and requires further investigation. Currently, there have been many studies into why *P. euphratica* has outstanding performance in salt tolerance compared with the other salt-sensitive poplar species. As a saline, desert living poplar specie, *P. euphratica* have developed a number of physiological and morphological mechanisms to acclimate to the salt condition, such as apoplastic and vacuolar salt accumulation, maintenance of cytosolic Na^+^/K^+^ homeostasis and development of leaf succulence [8]. Many genes involved in regulating Na^+^ transport and Na^+^/K^+^ homeostasis were cloned, including two Na^+^/H^+^ antiporter genes (*PeSOS1* and *PeNhaD1*) [9,10] and a NAC transcription factor (*PeNAC1*) [11]. It was also found that H+-ATPase provide the driving force for Na^+^/H^+^ antiporters to rebuild Na^+^ homeostasis in the cytosol [12,13]. Nevertheless, the functional roles of the vast majority of salt-responsive genes in *P. euphratica* remain uncharacterized. Among those that have been validated, the heat-shock transcription factor PeHSF directly binds heat-shock elements in the PeWRKY1 promoter to activate its expression and thereby enhance salt tolerance in *P. euphratica* [14]. Moreover, a putative xyloglucan endotransglucosylase/hydrolase gene (*PeXTH*) from *P. euphratica* was discovered to enhance salt tolerance by the development of succulent leaves in tobacco plants [15].

With the advent of high-throughput sequencing era, researchers can explore the salt tolerance mechanisms of *P. euphratica* from the perspective of genome and transcriptome on a large scale [16]. Li et al. identified 211 conserved and 162 novel miRNAs in *P. euphratica*, finding 15 miRNA-target pairs displayed reverse changing pattern under salt stress [16]. Qiu et al. assembled the *P. euphratica* transcriptome using the de novo assembly method and found some differentially expressed genes (DEGs) involved in ABA biosynthesis under salt condition [17]. A total of 34,279 protein-coding genes were identified in the *P. euphratica* genome and 57 positive selective genes (PSGs) were discovered, considerably larger than 45 PSGs in salt-sensitive black cottonwood (*P. trichocarpa*), including genes related to the regulation of ion homeostasis and scavenging of Reactive Oxygen Species [18]. Therefore, functional genomics studies in *P. euphratica* provide an important foundation for further understanding salt tolerance mechanisms in woody species.

As sessile organisms, plants cannot run away from the many threatening abiotic and biotic stresses and therefore need special mechanisms of stress avoidance and stress adaptation [2]. Plant’s stress response can be divided into four phases, including response phase, restitution phase, end phase and regeneration phase [19]. These phases involve preventing or alleviating damage, re-establishing homeostatic conditions in the stress environment and recovering growth, albeit at a reduced rate [1], thus it is very crucial to understand the complicated and dynamic biological process based on time-series experiments. Previous reports have shown that there are significant differences in physiological and biochemical status when *P. euphratica* seedlings were exposed to ST and LT salt stress [20]. However, most studies to date either fail to distinguish between short-term (ST) and long-term (LT) responses in *P. euphratica* or focus solely on the ST response, thereby overlooking LT adaptive mechanisms. In this paper, time-series gene expression profiles under ST and LT salt stress were monitored to comprehensively explore the salt resistance mechanism of *P. euphratica*.

To better understand how plants respond to environmental cues such as drought, salinity, and pathogens, researchers have employed gene regulatory networks to capture the complex interplay among transcription factors, regulatory genes, and target genes that orchestrate plant development, stress responses, and other physiological processes. Several computational approaches have been developed for time-series GRNs analysis in plant systems, including Bayesian networks [21], Gaussian graphical models and WGCNA [22]. However, these methods are typically either weighted and undirected or directed and unweighted, which limits their ability to capture the full complexity of regulatory network structures. The idopNetwork framework addresses these limitations by integrating quasi-dynamic ODEs, making it well-suited for analyzing the time-resolved transcriptional responses of plants under environmental stresses such as salinity. As a framework for dissecting and understanding the intricacies of complex systems, idopNetwork has been applied in gut microbiota [23], vaginal microbiota [24], cancer genomics [25], tumor-microenvironment interactions [26], and combinational therapies analysis [27]. To characterize the intricate dynamic GRNs and better understand the salt resistance mechanisms, we introduced this framework to analyze the time-series transcriptome data of *P. euphratica* roots under the salt condition.

The main goal of this study was to uncover the dynamic regulatory relationships among salt-responsive genes in *P. euphratica* using a systems biology approach. We hypothesized that salt stress triggers coordinated and time-dependent gene expression changes that can be captured by the idopNetwork framework, allowing the identification of key regulators and modules involved in salt tolerance. A total of 452 genes were identified as differentially expressed genes using DEseq2. The DEGs were functionally clustered, and then used to construct inter- and intra-module networks based on the quasi-dynamic ordinary differential equation model [28]. Gene regulatory network analysis discovered “hub” genes such as *XM_011048240.1* and *XM_011018353.1* and showed some interesting regulatory relationships between them. The potential functions of these genes were annotated to include plasma membrane, membrane and transmembrane transport. These results provide important insights into the molecular mechanisms of salt stress response in *Populus euphratica* and provide candidates for future experiments.

## 2. Results

### 2.1. Transcriptome Sequencing and Assembly

To investigate the transcriptomic response of *P. euphratica* to salt stress under short-term and long-term treatments, cDNA libraries were prepared from 18 root samples. These included three biological replicates at one time points for the control group and five time points for the salt treatment group. Sequencing was performed using the Illumina HiSeq2000 platform. Table 1 provides an overview of the sequencing and mapping results. On average, 129,818,256 reads were generated per sample, with a Q30 percentage of no less than 91.49%. These results indicate that both the throughput and sequencing quality were sufficiently high for subsequent analysis.

### 2.2. Analysis of Temporal DEGs

Differentially expressed genes were analyzed for mRNA expression at five time points under salt stress, including 0 h as the control (Figure 1A–E). Each time point contained 1278, 1225, 397, 1334, and 1452 significantly differentially expressed genes, respectively, which were upregulated and downregulated. A total of 10,581 upregulated and 11,335 downregulated genes were identified, respectively (Figure 1G). The number of DEGs at different times is shown in Figure 1H. Subsequently, a Venn diagram was generated to depict the intersection of DEGs across these five time points (Figure 1F), revealing 452 mRNAs with common differential expressions, which were selected for further analysis.

### 2.3. Gene Ontology (GO) Enrichment and KEGG Analysis of DEGs

To determine the functions of DEGs under salt stress at different time points, the biological pathways of DEGs in the roots of P. euphratica under salt stress at different times were annotated (Figure 2A–E). The analysis classified DEGs into corresponding functional categories. In the biological process (BP) category, the most significantly enriched processes under short-term (1 h, 6 h, 24 h) stress were regulation of DNA-templated transcription (GO:0006355), protein phosphorylation (GO:0006468) and transmembrane transport (GO:0055085), while under long-term (7 d, 18 d) stress in addition to the above functions, there was also the ethylene-activated signaling pathway (GO:0009873). For cellular components (CC), short-term stress mainly enriched nucleus (GO:0005634) and plasma membrane (GO:0005886), and long-term stress also enriched membrane (GO:0016020) and extracellular region (GO:0005576). In the molecular function (MF) category, the top four enriched terms under short-term influence were ATP binding (GO:0005524), DNA binding (GO:0003677), DNA-binding transcription factor activity (GO:0003700), and heme binding (GO:0020037). The main function enriched under long-term influence was protein kinase activity (GO:0004672).

Except for GO analysis, we also performed KEGG pathway enrichment analysis on DEGs under different salt stress durations and displayed the results in the form of bubble charts (Figure 2F–J). Under salt stress at the first four points (1 h, 6 h, 24 h, and 7 d), the main enriched metabolic pathways included metabolic pathways (peu01100), Biosynthesis of secondary metabolites (peu01110), Plant hormone signal transduction (peu04075) and MAPK signaling pathway-plant (peu04016). After 18 days of salt stress treatment, plant–pathogen interaction (peu04626) increased in the significantly enriched metabolic pathway entries.

### 2.4. Functional Cluster of Temporal DEGs

Functional clustering was conducted on the 452 DEGs identified from the previous analysis. Here, we define the Expression Index as the aggregated sum of the expression levels of all DEGs, which could serve as a robust predictor to distinguish among different treatments [29]. These DEGs were then clustered into nine distinct modules (Figure 3A), each exhibiting unique expression trends. For instance, genes in modules 1, 3, 5, and 7 displayed an upward trend as the expression index increased, while those in modules 2, 6, 8, and 9 showed a downward trend. Module 4, which contained the largest number of genes, exhibited a relatively stable expression pattern. The optimal number of clusters was determined using the Bayesian Information Criterion (BIC) (Figure 3B).

### 2.5. idopNetwork Reconstruction for DEGs

A two-layer idopNetwork was constructed using the 452 differentially expressed genes (Figure 4A). The mean vectors of the nine modules obtained through functional clustering were used to build a coarse-grained network, illustrating interactions between modules across different time points. The network structure revealed module M4 as the hub module, displaying the highest number of interactions with other modules. This module contained the largest number of genes, with a total of 341. Subsequently, a fine-grained (gene-level) network was constructed for all genes within module M4, and the Table A2 shows the core genes of the module at each time point. The topological structure and regulatory dynamics of the network exhibited distinct changes over time. During short-term salt stress (0 to 24 h), most genes in the network exhibited a promoting effect. However, after prolonged salt stress (7 days), idopNetwork analysis indicates gene interactions become more inhibited. Under extended salt stress (18 days), the interactions returned to a promoting state. Additionally, the in-degree and out-degree of each gene in the fine-grained network within module M4 were analyzed, and the top 50 genes with the most interactions were highlighted (Figure 4B).

Functional enrichment analysis of genes in module 4 (Figure 5) revealed key enriched terms across different GO categories. In the molecular function (MF) category, the most highly enriched terms were heme binding (GO:0020037) and signaling receptor activity (GO:0038023). Within the biological process (BP) category, the most significantly enriched processes were transmembrane transport (GO:0055085) and signal transduction (GO:0007165). For the cellular component (CC) category, the predominant terms included plasma membrane (GO:0005886) and membrane (GO:0016020).

A distinctive feature of idopNetwork is that it models a gene’s expression as comprising two components: its independent expression and the regulatory effects (dependent expression) exerted by its interacting genes [28,30]. We decomposed the effects of core genes, positing that the actual expression of a gene is influenced not only by its own independent expression level but also by the regulatory effects of other genes, whether promoting or inhibiting. Figure 6 presents the decomposition of the effect curve, with two representative genes, XM_01101835

3.1 and XM_011037175.1, highlighted for illustration. The effect curve provides a detailed depiction of the gene’s specific expression pattern. Taking gene XM_011018353.1 as an example, its expression steadily increases as the salt stress time extends, accompanied by a growing promoting effect from other genes. A similar pattern was observed in gene XM_011037175.1. Network analysis further revealed that the expression levels of genes XM_011016547.1 and XM_011027700.1, which are connected to XM_011018353.1, also increased significantly over time under salt stress, driven by the promoting influence of XM_011018353.1. This finding reinforces the trend of mutual promotion among genes in the network, highlighting how the synergistic effects between genes become more pronounced under salt stress conditions.

### 2.6. Validation of RNA-Seq Data by RT-qPCR

To validate the sequencing results and investigate the dynamic expression patterns of mRNAs under salt stress, six differentially expressed mRNAs were randomly selected for quantitative real-time PCR (RT-qPCR) analysis (Figure 7). The error bars represent standard errors, and one-way ANOVA with Duncan’s post hoc test was used to analyze mean significance between samples. Despite slight differences in expression levels, the RT-qPCR data showed expression patterns consistent with the RNA-seq results, confirming the reliability of RNA-seq and indicating that these differentially expressed mRNAs are involved in *P. euphratica*’s response to salt stress. The correlation between RT-qPCR and RNA-seq expression data was assessed using Pearson correlation analysis, further supporting the consistency between the two datasets (Figure A1). For example, *XM_011030499.1* showed a rapid upregulation under short-term salt stress, reaching its peak at 6 h, followed by a decline under long-term stress, while *XM_011030272.1* showed downregulation under short-term stress and upregulation under long-term stress. These findings, consistent with the RNA-seq results, suggest that the genes exhibit time-dependent dynamic expression patterns. Specifically, *XM_011006869.1* and *XM_011039716.1* showed significant upregulation at 1 h, followed by a gradual decrease, indicating a short-term salt stress response. Meanwhile, *XM_011030499.1* and *XM_011041653.1* displayed low expression at 0 h, followed by a continuous increase within 6 h of salt stress and then a decrease. Lastly, *XM_011030272.1* and *XM_011043514.1* showed little change from 0 to 6 h but were significantly upregulated after 24 h, suggesting a long-term salt stress response.

## 3. Discussion

*P. euphratica*, as an arbor species capable of surviving in arid and saline alkali desert area in Northwest China, is known as a perfect representative for studying into stress resistance in woody species. In recent years, researchers comprehensively investigated the mechanisms of salt tolerance in *P. euphratica* from the perspectives of physiology, biochemistry, and molecular biology [5,31,32]. As the rapid development of the next-generation sequencing technology, it is possible to monitor the genome-wide gene expression profiles, which is conducive to study the gene regulatory mechanisms of salt tolerance in *P. euphratica* [16,17]. Here, we used time-series experimental design and preliminarily parsed the gene expression patterns of the *P. euphratica* roots under salt condition using high-throughput RNA-seq technology, providing a clear direction for further precise analysis and functional verification.

Salt stress, a significant environmental stressor, broadly impacts plant growth and is a major factor limiting plant development in arid regions worldwide [33]. Recent studies indicate that salt stress substantially impairs root growth and function. Under saline conditions, root elongation slows, with increasing salt concentrations correlating with shorter root lengths [34,35]. These phenomena are observed in our experiments. Additionally, salt stress alters the uptake and distribution of essential ions, such as sodium (Na^+^) and potassium (K^+^), leading to ion imbalances that impair root function and disturb the plant’s overall nutritional equilibrium, resulting in stunted growth and reduced vigor [36]. As Na^+^ levels rise, fluctuations in K^+^ concentrations further disrupt critical processes, including root water uptake and nutrient transport. These physiological changes in roots can have far-reaching effects on plant growth and survival under saline conditions, manifesting as delayed root initiation, altered root architecture, and diminished water and nutrient uptake efficiency [37]. Based on this, we designed salt stress experiments at different time points to try to explain the response of gene expression to the duration of salt stress.

RNA-seq is useful for biologists to comprehensively understanding the fluctuations of gene expression levels in the processes of biological development and adaptation to the changing environments [38]. However, it is an extremely complicated process involving reverse transcription, amplification, fragmentation, purification, adaptor ligation and sequencing. Improper operations, transcriptome complexity, and RNA-seq intrinsic biases (such as GC bias and nucleotide composition bias) can make the data imperfect [39]. Therefore, comprehensive quality assessment is the first and most pivotal step for all downstream analyses. In this study, the sequencing error rate was less than 0.02%, the average GC content was about 43%, and Q30 was higher than 92% in all samples. All the assessment results showed that the sequencing data in this study were highly reliable. Root tissues often exhibit early and robust transcriptomic changes in response to external stimuli, making them an ideal system to capture initial molecular events [40]. Although the transcriptomic landscape of young plant roots may differ substantially from that of mature plants, the findings of this study primarily reflect the gene expression patterns typical of young, actively growing root tissues. Future studies comparing different developmental stages would be valuable to fully understand the dynamic transcriptional regulation across plant maturation.

The number and characteristics of DEGs at different time points exhibited a clear time dependence, indicating that *P. euphratica* employs distinct molecular response mechanisms during the early and late stages of salt stress. GO enrichment and KEGG pathway analyses of the differential genes revealed significant enrichment in multiple pathways closely associated with salt stress tolerance. In the early stages of stress, gene expression primarily concentrates on pathways related to DNA transcription templates, protein phosphorylation, and transmembrane transport, suggesting that *P. euphratica* rapidly activates defense mechanisms to maintain cellular homeostasis. As stress persists, gene expression at later time points increasingly focuses on ethylene-activated signaling pathways and plant hormone signal transduction, indicating a gradual transition from initial defense mechanisms to more sustained adaptive mechanisms. The identification of these genes and pathways offers new insights into the physiological adaptability of *P. euphratica* in response to salt stress environments.

As a versatile framework for describing biomolecular interactions, network representations have been widely applied across animal, plant, and microbial systems; examples include human gene interaction network [41], yeast protein interaction network, and intestinal metabolite interaction network [42,43]. In this study, we introduced a new network construction method, idopNetwork, and successfully constructed a multi-level, directional, and weighted interaction network between DEGs, which is different from previous bioinformatics methods. The core concept of the idopNetwork method is to introduce evolutionary game theory to decompose the actual expression level of a gene into its own independent expression component and the combined effects from other genes. Thus, the observed expression value can be resolved by solving differential equations. The influence from other genes can be either positive (promotion) or negative (inhibition), and these effects dynamically change over time and under different environmental conditions. As gene expression levels shift across different time points under salt stress, the regulatory influences from other genes also undergo corresponding changes. Capturing these dynamic variations is crucial for understanding the regulatory mechanisms of genes under both short-term and long-term salt stress. By constructing such networks, we found the hub module M4, which contains the most genes and is most closely connected to other modules. It is speculated that it plays a central regulatory role in the salt stress response of *P. euphratica*. Further analysis of the gene interaction relationship within module 4 at different periods found that under short-term stress, the network showed a trend of mutual promotion. However, when the stress lasted until the seventh day, this trend turned into a large amount of mutual inhibition. This dynamic change of the gene interaction network reflects the gradual process of *P. euphratica* from early rapid stress response to long-term adaptation. Through in-depth network construction and node degree statistical analysis of this hub module, we found hub genes with topological structures similar to core modules at the gene network level. These genes play a vital regulatory role in the network.

For example, the hub gene *XM_011048240.1* exhibits a high level of interaction with other genes, forming 15 connections with neighboring nodes. These include 8 promoting outgoing edges, 1 inhibiting outgoing edge, and 6 promoting incoming edges, suggesting its significant regulatory role. To further investigate the potential function of the *P. euphratica* gene *XM_011048240.1*, a homology search was performed against the *Arabidopsis thaliana* protein database using BLASTp (v2.16.0). The analysis revealed that *XM_011048240.1* shares a high degree of sequence similarity with the Arabidopsis protein *NP_178407.2*, corresponding to the gene *AT2G03070*. This gene encodes transcript variant X6 of RNA polymerase II transcription mediator complex subunit 8 (MED8), which is involved in transcriptional regulation. In *Arabidopsis thaliana*, MED8 has been shown to regulate flowering time and is also implicated in cell wall composition and cell elongation [44]. These findings underscore the essential role of MED8 in plant growth, development, and environmental adaptation. This suggests that *XM_011048240.1* may play a similar role in coordinating developmental processes and stress-related signaling pathways in *P. euphratica*, potentially linking environmental adaptation to growth regulation. Furthermore, our results support the transcriptional regulatory function of *XM_011048240.1*, suggesting that it may play a crucial role in the adaptation of *Populus euphratica* to salt stress. By constructing idopNetwork containing all DEGs and analyzing the quantitative interaction relationship of differentially expressed genes, this regulatory network provides an important reference for identifying and predicting gene regulatory relationships.

While this study provides valuable insights into the dynamic changes in gene expression in *P. euphratica* in response to salt stress, several limitations need to be addressed in future research. First, the current study focused primarily on the transcriptomic level, whereas integrating proteomic and metabolomic analyses could provide a more complete understanding of the molecular mechanisms underlying salt tolerance in *P. euphratica*. Furthermore, while *P. euphratica* is an excellent model for studying salt stress, comparisons with other salt-tolerant species, such as *P. trichocarpa*, are necessary to better understand species-specific adaptations. In future work, it would be valuable to conduct comparative transcriptomic and network analyses between *P. euphratica* and *P. trichocarpa* to identify conserved and species-specific pathways for salt tolerance [45,46]. Nevertheless, future work may also include a systematic comparison between idopNetwork and other widely used network inference methods, such as WGCNA and GRNBoost2, to further evaluate their relative strengths and complementarity in identifying key regulatory modules.

## 4. Materials and Methods

### 4.1. Plant Materials and Salt Stress Treatment

*P. euphratica* samples were collected from the central Inner Mongolia, China. After confirming that they were genotype-identical using microsatellite markers, a healthy sampling was selected and cloned via tissue culture to produce multiple young clones. These clones were then transplanted into individual pots containing pearlite and vermiculite and placed at a greenhouse with consistent temperature (22 °C) and illumination conditions (16 h photo period) at Beijing Forestry University. After three months, 24 uniformly growing seedlings were selected for the following salt stress treatment. They were randomly divided into two groups. One group was treated with 300 mM NaCl solution based on preliminary experiments [47], while the other group was treated with equal amount of water and used as control. The root samples of these seedlings were collected at 0 h, 1 h, 6 h, 24 h, 7 d, and 18 d after the salt stress treatment. Meanwhile, the root samples in the control group were collected at 24 h and 7 d. At each time point, three seedlings were selected as biological replicates. After collecting, all the samples were immediately frozen in liquid nitrogen for RNA extraction. The first sample collected at 24 h and 7 d in the control group was named as eCR_24h1 and eCR_7d1, respectively, whereas the first sample collected at 24 h in the salt-treated group named as eSR_24h1, and so forth.

### 4.2. RNA Extraction and High-Throughput Sequencing

All the root samples were used for RNA extraction using the RNAprep Pure Plant Kit (Polysaccharides & Polyphenolics-rich) (TIANGEN, Beijing, China). RNA integrity and concentration were measured using the RNA Nano6000 Assay Kit of the Bioanalyzer 2100 system (Agilent Technologies, Santa Clara, CA, USA) and Qubit^®^ RNA Assay Kit in Qubit^®^ 2.0 Fluorimeter (Life Technologies, CA, USA), respectively. The construction and sequencing of RNA-seq libraries were performed at Novogene Bioinformatics Technology Co., Ltd. (Beijing, China). Using the rRNA-depleted RNA by NEBNext^®^ Ultra^TM^ Directional RNA Library Prep Kit for Illumina^®^ (New England Biolabs, Ipswich, MA, USA), nearly 3 mg RNA per sample was used for library preparation according to the manufacturer’s recommendations. After that, all sequencing libraries were submitted for 100 bp paired-end sequencing on an Illumina Hiseq 2000 platform (Table A1).

### 4.3. Reads Mapping and Analysis

After the raw reads were generated, we used Trimmomatic software v0.32 (ILLUMINACLIP:TruSeq3-PE.fa:2:30:10 SLIDINGWINDOW:4:15 MINLEN:36) to trim the adaptor sequences and then used FastQC software v0.10.0 to check the quality of the trimmed reads [48,49]. The resulting reads were mapped to the *P. euphratica* genome using TopHat 2.0 program [50]. Then, we assembled the mapped reads using Cufflink program, which was run with ‘min-frags-per-transfrag = 0’ and ‘-library-type’ and with other parameters set as default [51]. After assembly, gene expression levels were calculated using Cuffdiff (v2.1.1) and reported as FPKM values.

### 4.4. Identification of Differentially Expressed Genes and GO Analysis

To identify short-term and long-term salt-responsive genes, we provided an elaborate experimental design containing different times, including six time points under the salt stress treatment, each with three biological replicates and used the average expression values across replicates for subsequent analysis. As a result, we adopted a strategy for identifying differentially expressed genes (DEGs). Consider the time factor for RNA populations under salt stress and then observe changes in gene expression and their interactions at different time points. RNA-seq data were initially processed as FPKM values, normalized by scaling the total expression of each sample to one million, and converted into pseudo-counts by rounding for DESeq2 (v1.42.1) analysis. Genes with low expression (total counts ≤ 10,000) were filtered out. Differential expression analysis was conducted using the internal median-of-ratios normalization method, with model fitting set to fitType = “mean”. We performed differential gene expression analysis under salt stress at 0 h, 1 h, 6 h, 24 h, 7 d, and 18 d using the DESeq2 R package (v1.42.1) [52,53]. After adjusting for false discovery rate (FDR) using the Benjamini–Hochberg method, genes with a |log2FoldChange| ≥ 1 and adjust *p* value ≤ 0.05 were classified as differentially expressed genes. No significant batch effects were detected during analysis; therefore, batch correction was not applied. GO enrichment analysis of DEGs was performed using David (https://david.ncifcrf.gov/ (accessed on 20 April 2025)) [54,55].

### 4.5. Clustering Analysis

The genome of *P. euphratica* encodes tens of thousands of mRNAs, which interact within a large-scale network. However, it is unrealistic to assume that all mRNAs interact with each other [56,57]. In this work, we introduced a method based on developmental modularity theory, called Functional Cluster to ensure network sparsity [28,58]. To explore the patterns of these gene expressions, we performed maximum likelihood estimation clustering based on a mixture model on the RNA-seq data. By clustering mRNAs into distinct modules based on similar expression patterns, we categorized them into functional groups, allowing us to reconstruct a multi-layer network that reflects these modular interactions.

### 4.6. Gene Regulatory Network Construction

The response to salt stress is a dynamic biological process, which is regulated by a sophisticated network composed of many genes and their products. Therefore, the construction of dynamic GRN is the key to understanding the mechanisms of salt tolerance in *P. euphratica*. Dong et al. proposed a model idopNetwork based on quasi-dynamic ordinary differential equations (qdODE) to capture the interaction network in the ecological community [28]. In contrast to the traditional WGCNA method, we used this network model to construct a dynamic GRN of DEGs in *P. euphratica* root system under salt stress at different time points. This model could flexibly deal with linear and nonlinear regulation effects. The visualization of GRNs was achieved using cytoscape3.9.1 software.

### 4.7. Quantitative Real-Time PCR (RT-qPCR)

Total RNA was extracted from roots using the RNAprep Pure Plant Kit (Polysaccharides & Polyphenolics-rich) (TIANGEN, Beijing, China). About 0.5 μg RNA was reverse-transcribed into first-strand cDNA using HiScript^®^ II Q RT SuperMix for qPCR (Vazyme). RT-qPCR was performed on a BIO-RAD CFX ConnectTM machine by using KAPA SYBR^®^ FAST qPCR Kit Master Mix(2X) Universal (KAPA BIOSYSTEMS). Gene-specific primers and internal control primers, which were designed by Primer BLAST v2.11.0, were shown in Table A1. All RT-qPCR reactions were carried out in three biological and three technical replicates with 57 °C annealing temperature and 40 amplification cycles. The relative expression levels were calculated by the 2^−ΔΔc^ (t) method.

## 5. Conclusions

This study revealed the dynamic changes in the gene expression network of *P. euphratica* in response to salt stress across different time points, providing new insights into the molecular mechanisms underlying salt tolerance. These findings not only shed light on the adaptive evolutionary strategies of *P. euphratica* but also lay a solid foundation for understanding its responses to saline environments. Although our results substantially advance current knowledge, further investigations integrating multi-omics approaches and cross-species comparisons are needed to fully elucidate the complex regulatory networks involved in salt tolerance. Such future research will not only deepen our understanding of the adaptive strategies of *P. euphratica* but also support breeding programs aimed at developing more salt-tolerant plants for agricultural and ecological applications.

## Figures and Tables

**Figure 1 ijms-26-04091-f001:**
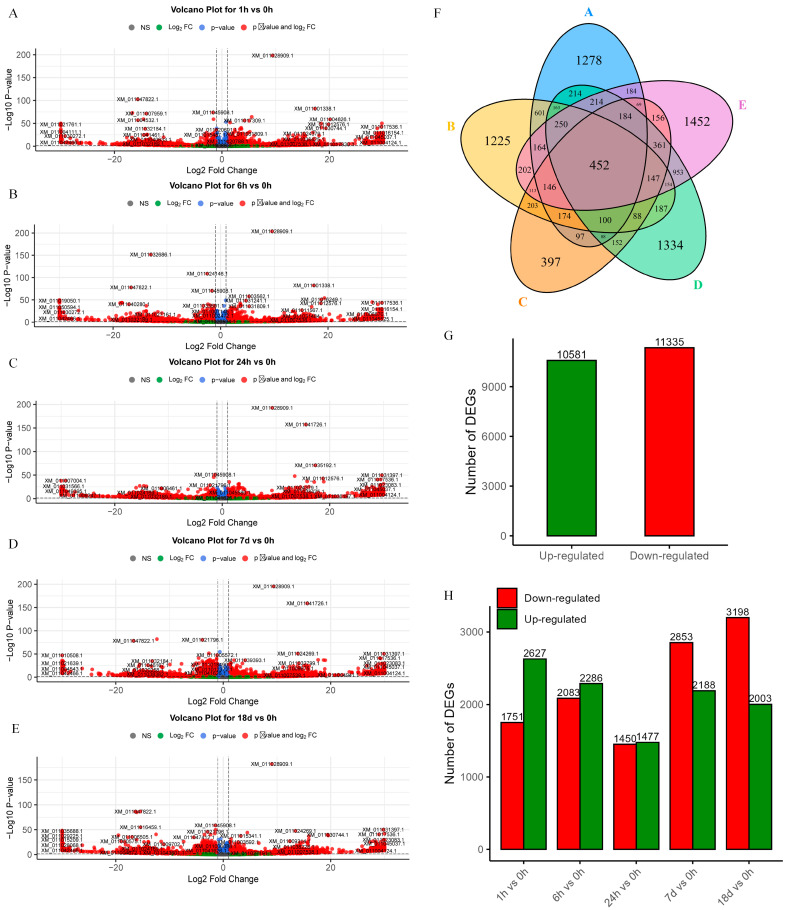
DEGs volcano plots at five different time points and statistic information. (**A**–**E**) Volcano plot of DEGs for 1 h/0 h, 6 h/0 h, 24 h/0 h, 7 d/0 h, and 18 d/0 h. (**F**) Venn diagram of DEGs at five time points, from A to E is 1 h/0 h, 6 h/0 h, 24 h/0 h, 7 d/0 h, and 18 d/0 h. (**G**) Total number of upregulated and downregulated genes. (**H**) Number of upregulated and downregulated genes at five different time points.

**Figure 2 ijms-26-04091-f002:**
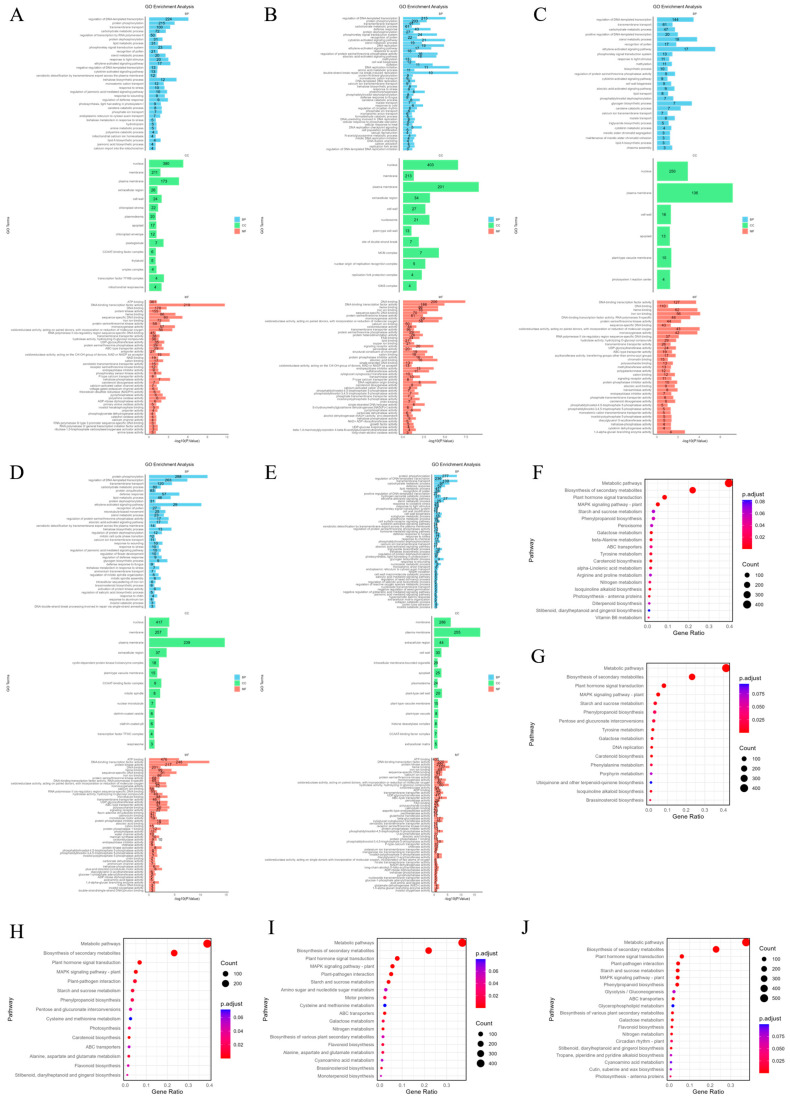
GO and KEGG annotation of DEGs. (**A**–**E**) GO annotation distribution of DEGs in *P. euphratica* at different time points under salt stress. (**F**–**J**) KEGG pathway analysis of DEGs in *P. euphratica* at different time points under salt stress.

**Figure 3 ijms-26-04091-f003:**
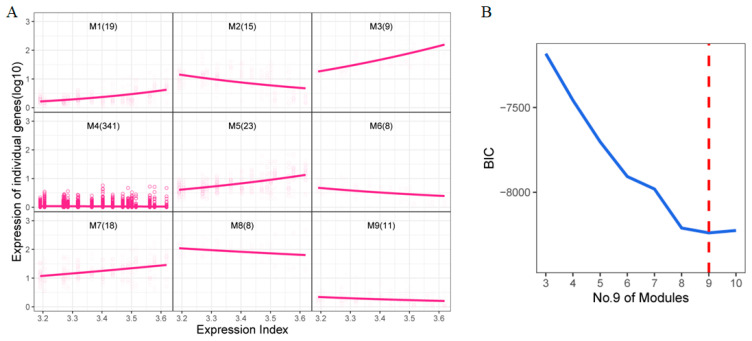
Functional clustering of DEGs (**A**) Functional clustering of 452 DEGs. The horizontal axis is the total expression index, and the vertical axis is the expression of an individual gene (log10). (**B**) The value of BIC. The dashed red line shows 9 is the best cluster number.

**Figure 4 ijms-26-04091-f004:**
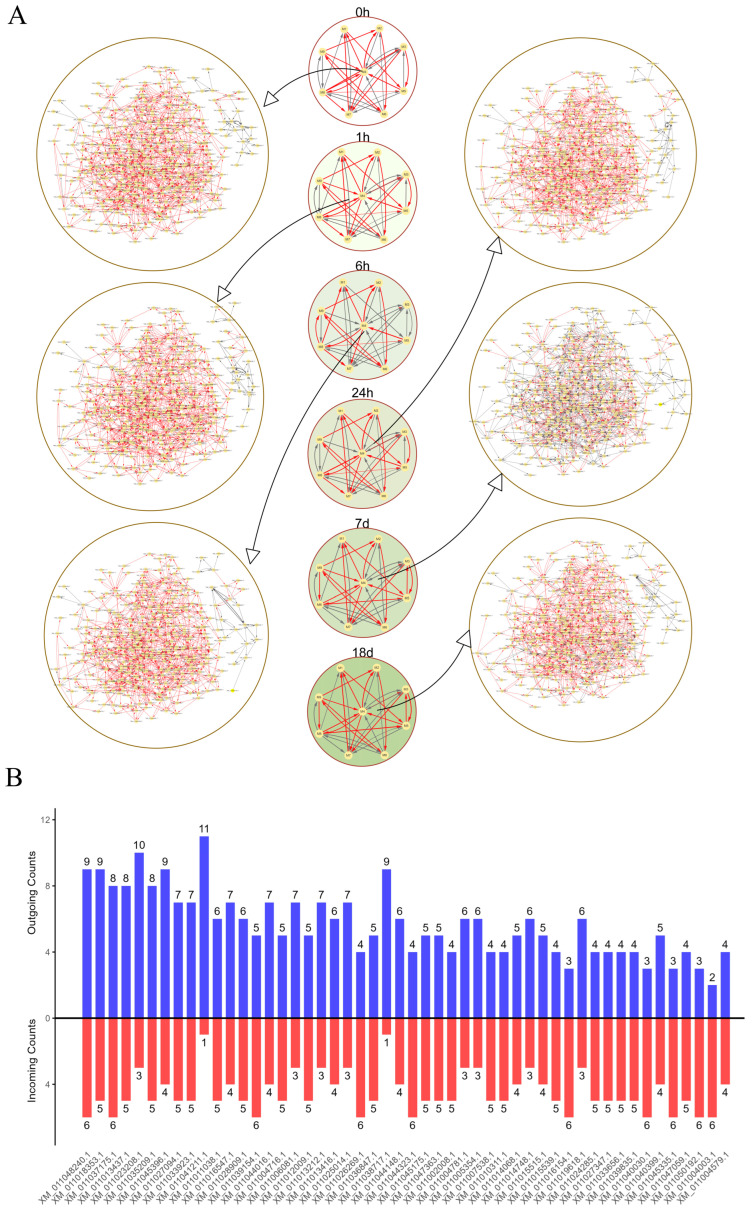
idopNetwork and nodes degree among hub modules. (**A**) Two-layer idopNetwork at different time points. Each node represents a module or an individual gene, and the edge represents the interaction between nodes. Red represents promotion and gray represents inhibition. (**B**) Incoming and outgoing node degrees of the top 50 genes in gene-level networks.

**Figure 5 ijms-26-04091-f005:**
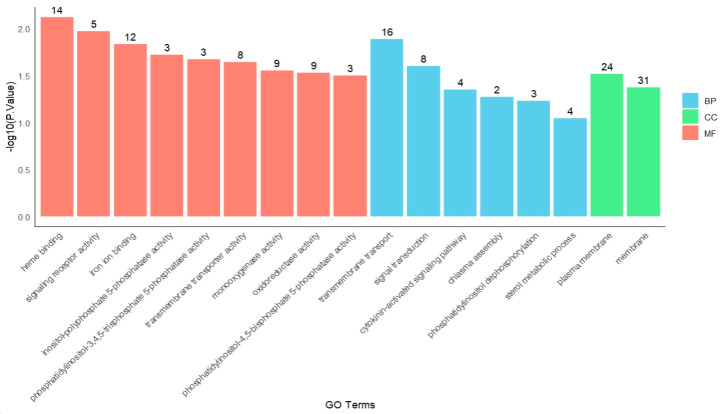
GO enrichment of module 4.

**Figure 6 ijms-26-04091-f006:**
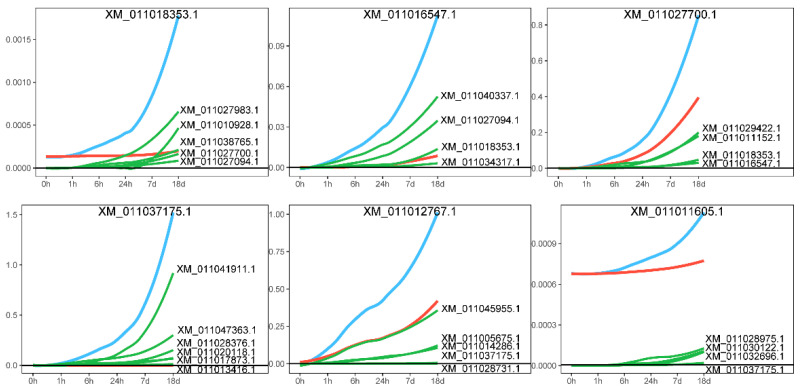
Effect curve decomposition diagram. The blue line represents the actual expression effect of the gene, the red line indicates the independent expression effect of the gene itself, and the green line shows the dependent expression exerted by other genes on it.

**Figure 7 ijms-26-04091-f007:**
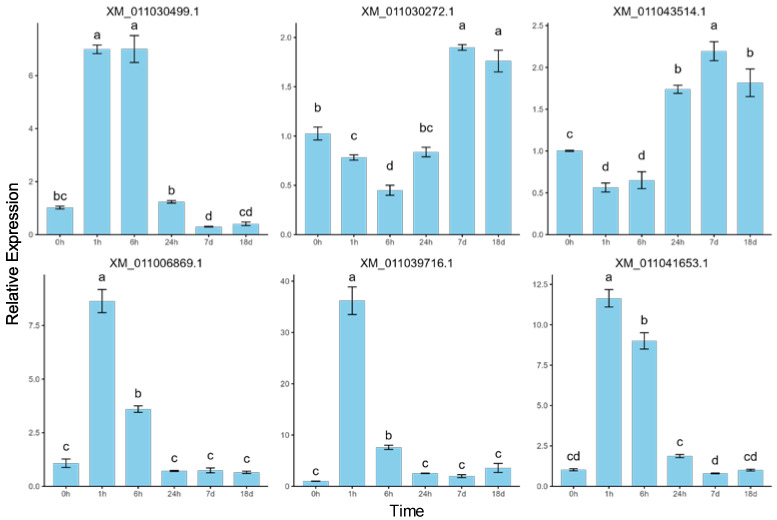
RT-qPCR validation of the expression patterns of six randomly selected differentially expressed mRNAs. The same letters above bars indicate no significant difference between samples (Duncan’s test, *p* < 0.05).

**Table 1 ijms-26-04091-t001:** Overview of the sequencing and mapping result.

Samples	Raw Reads	Clean Reads	Clean Bases	Q30 (%)	GC Content (%)
eSR_0h1	120,061,832	117,037,376	17.56G	93.42	43.91
eSR_0h2	126,544,958	123,009,778	18.45G	93.24	43.74
eSR_0h3	127,970,524	124,812,530	18.72G	93.59	43.41
eSR_1h1	123,483,826	120,287,176	18.04G	94.23	44.15
eSR_1h2	108,341,308	105,816,148	15.87G	94.21	44.66
eSR_1h3	126,160,386	122,993,670	18.45G	94.10	44.35
eSR_6h1	131,378,764	124,558,814	18.68G	94.36	45.28
eSR_6h2	120,654,756	117,432,120	17.61G	93.86	44.31
eSR_6h3	144,094,448	139,794,392	20.97G	94.19	45.05
eSR_24h1	153,980,886	148,217,346	22.23G	92.62	42.85
eSR_24h2	140,975,854	135,396,226	20.31G	92.60	43.29
eSR_24h3	159,602,918	153,235,106	22.99G	92.65	42.98
eSR_7d1	115,600,962	111,504,632	16.73G	91.49	43.63
eSR_7d2	142,277,696	137,337,968	20.6G	91.56	43.79
eSR_7d3	146,228,598	141,376,682	21.21G	91.80	43.45
eSR_18d1	128,754,356	125,175,170	18.78G	93.80	44.47
eSR_18d2	131,503,632	127,914,480	19.19G	93.65	43.96
eSR_18d3	114,477,542	111,408,754	16.71G	94.04	43.96

## Data Availability

The data presented in this study are available on request from the corresponding author.

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
