# Peer review of "idopNetwork Analysis of Salt-Responsive Transcriptomes Reveals Hub Regulatory Modules and Genes in Populus euphratica"

_ijms, 2025, doi:10.3390/ijms26094091_

Round 1
Reviewer 1 Report
Comments and Suggestions for Authors
In the current manuscript, the authors performed RNA-seq analyses on salt stress-treated Populus euphratica root samples over a time-course, and identified gene clusters with distinct temporal regulations in the context of prolonged salt conditioning. The experiments were carefully conducted, the data were properly analyzed and presented and the paper generally well-written. The work may be very helpful to colleagues in the field studying the impacts of stress conditions to plants. I have a few suggestions.
- It would be helpful if the authors could elaborate on the choice of using 300mM NaCl as the sole salt stress condition tested in this work. It may be beneficial, for instance, to cite works reporting physiologically relevant soil conditions for the Populus euphratica plant to justify adopting this single condition.
- Please add some discussions to explain as to why only total RNA from the root tissues of the plants were analyzed; and perhaps from a high level, discuss and/or acknowledge the potentially distinct transcriptomic backgrounds those young plants (and their root tissues, as used in this study) may be compared to adult plants.
- Figures 2 and 4 are extremely busy, it may be beneficial to re-organize them and/or moving certain contents to supplementary files.
- Lines 211-212:, “XM_011018353.1, also increased significantly over time under salt stress, driven by the promoting influence of XM_011018353.1”: since there’re no direct evidence establishing causal relationship between the two genes, it appears improper to postulate such.
- The RT-qPCR data in Figure 7 was generated in the hope to validate the RNA-seq data, as such, please also show the correlation of the two datasets.
Minor:
- Line 86, “exploring”: change to “explore”.
- Figure 1 A-E: the volcano plots appear overly flattened, it may be helpful to isolate the outliers on the y-axis and stretch the rest of the datapoints.
Reviewer 2 Report
Comments and Suggestions for Authors
The article is written correctly. However, it contains some errors that should be corrected.
The introduction refers to the content of the work, but does not include the main goal of the work
and the research hypothesis, which should be added. The results are written correctly and legibly.
The discussion refers to the latest research on the discussed problem. The research methods and analysis of the results were used correctly.
However, a summary is missing. The authors presented it at the end of the discussion.
However, they should write a separate chapter at the end of the article and present the results in detail.
After correcting the errors, the work can be published in Int. J.Mol. Sci.
Reviewer 3 Report
Comments and Suggestions for Authors
I appreciate the opportunity to review the manuscript entitled "idopNetwork analysis of salt-responsive transcriptomes reveals hub regulatory modules and genes in Populus euphratica." This study explores the temporal dynamics of gene expression in P. euphratica under salt stress using RNA-seq data and a novel network inference method called idopNetwork. The authors integrate time-course transcriptomics, differential gene expression analysis, and regulatory network modeling to identify key regulatory modules and hub genes potentially involved in salt stress response.
The manuscript presents a well-organized experimental design, with clear treatment groups, high-quality transcriptome data, and appropriate time points to capture dynamic transcriptional changes. One of the key strengths of this study is the application of the idopNetwork model, which represents an innovative approach to infer regulatory relationships and identify gene modules based on information topology and dynamic expression profiles. The validation of RNA-seq data through RT-qPCR further strengthens the reliability of the dataset.
The work is timely, relevant, and contributes to the growing field of plant stress genomics by addressing an important question: how transcriptional networks evolve under prolonged abiotic stress in a stress-tolerant tree species.
However, I have a few concerns and suggestions that, if addressed, could significantly improve the clarity, interpretability, and impact of the manuscript:
Major Comments
- Lack of comparison with standard network methods
While the use of idopNetwork is innovative, the manuscript would greatly benefit from a comparison with more widely adopted network analysis methods, such as WGCNA or GRNBoost2. This would provide context and help validate the novel method by showing its advantages or complementarity in identifying key regulatory modules or hub genes. - Methodological details in DEG analysis
The differential expression analysis lacks some methodological transparency. For example, the manuscript does not clearly state: - How normalization of expression data was handled (e.g., TPM, FPKM, or raw counts with DESeq2 normalization).
- Whether any batch effects were present and, if so, how they were accounted for.
- The number of biological replicates used per time point and treatment.
Including this information is crucial for reproducibility and to evaluate the robustness of the differential expression results.
- Interpretation of the regulatory network structure
The manuscript introduces the concepts of expression being decomposed into independent and dependent components, as well as promoting and inhibiting interactions in the network. However, the biological interpretation of these features is not sufficiently clear. It would strengthen the manuscript to elaborate on what these inferred interactions imply in terms of gene regulation—particularly how these translate into physiological processes during salt stress. - Underexplored biological relevance of hub genes
The identification of hub genes such as XM_011048240.1 is an important outcome of the study. However, the manuscript would benefit from a deeper discussion of the functional relevance of these genes, especially those consistently identified across multiple time points or regulatory modules.
While we fully understand that functional validation in the lab can be resource-intensive and time-consuming, the authors could still improve this section by: - Performing additional in silico analyses (e.g., domain annotation, conserved motif identification, subcellular localization predictions).
- Comparing hub gene functions with previously known regulators in model systems such as Arabidopsis.
- Expanding the discussion on how these hub genes might mechanistically contribute to salt stress response in P. euphratica.
- Figures and data visualization
Some of the network figures could benefit from improved legends and clearer annotations (e.g., explaining node colors, edge weights, time point grouping). Consider adding a table summarizing the top hub genes per time point/module to aid interpretation.
Minor Comments
- The introduction is well-written, but could include a brief paragraph reviewing current methods for time-series network analysis in plant systems, to better contextualize the novelty of the idopNetwork model.
- The term “co-regulatory genes” is used frequently—clarify whether this refers to co-expressed genes, inferred upstream regulators, or both.
- Standardize figure and table references throughout the manuscript (e.g., "Figure 3" instead of "Fig.3").
- The discussion would be more impactful with a concluding paragraph summarizing the key biological insights and limitations.
Round 2
Reviewer 3 Report
Comments and Suggestions for Authors
I have carefully evaluated the revised version of the manuscript and the authors' point-by-point response to my previous comments. I am pleased to confirm that the authors have adequately addressed all my concerns: They have justified the use of the idopNetwork framework; the Introduction now includes a concise review of existing time-series network analysis approaches; the methodology section was significantly improved; the biological interpretation of the network features, has been expanded and clarified. I believe the authors have made substantial improvements and that the study offers valuable insights into gene regulatory dynamics under salt stress in P. euphratica.